# Triaxial Testing of Geosynthetics Reinforced Tailings with Different Reinforced Layers

**DOI:** 10.3390/ma13081943

**Published:** 2020-04-20

**Authors:** Fu Yi, Changbo Du

**Affiliations:** 1China Coal Research Institute, Beijing 100013, China; 2College of Civil Engineering, Liaoning Technical University, Fuxin 123000, China

**Keywords:** reinforced tailings, geosynthetics, triaxial test, shear strength, reinforcement effect

## Abstract

To evaluate the shear properties of geotextile-reinforced tailings, triaxial compression tests were performed on geogrids and geotextiles with zero, one, two, and four reinforced layers. The stress–strain characteristics and reinforcement effects of the reinforced tailings with different layers were analyzed. According to the test results, the geogrid stress–strain curves show hardening characteristics, whereas the geotextile stress–strain curves have strain-softening properties. With more reinforced layers, the hardening or softening characteristics become more prominent. We demonstrate that the stress–strain curves of geogrids and geotextile reinforced tailings under different reinforced layers can be fitted by the Duncan–Zhang model, which indicates that the pseudo-cohesion of shear strength index increases linearly whereas the friction angle remains primarily unchanged with the increase in reinforced layers. In addition, we observed that, although the strength of the reinforced tailings increases substantially, the reinforcement effect is more significant at a low confining pressure than at a high confining pressure. On the contrary, the triaxial specimen strength decreases with the increase in the number of reinforced layers. Our findings can provide valuable input toward the design and application of reinforced engineering.

## 1. Introduction

As geosynthetics can perform filtration, drainage, isolation, and protection, reinforced soil technology has been widely used in the fields of civil engineering reinforcement, protection, water conservation, highway, railway, and coal mine construction, and other engineering fields [1,2]. At present, considerable engineering experience has been accumulated in this field, and the theoretical research has been developed to a certain extent. However, few studies have been conducted on the application of reinforcement technology in tailings dams. The tailings pond is a kind of special industrial building, and the quality of its operation not only affects the economic benefits of a mining enterprise, but is also closely related to the life and property safety of the residents in the lower reaches of the reservoir area and the surrounding environment. To achieve a larger safety factor, the problems of small slope ratio and large land occupation are common in the construction of tailings ponds. There is no tailing pond that can be used in the natural superior terrain because of the shortage of favorable land use conditions, which seriously affects the tailing pond service life. Generally, the stability and accumulation height of the tailings dam are solved by strengthening the tailings dam. The main strengthening methods include the drainage consolidation method, vibro compaction method, grouting method, and geosynthetics reinforcement method. It is an effective method to use geosynthetics to improve the slope ratio of tailings dam [3,4]. In the process of tailing dam construction, a layer of geosynthetics is laid at a certain interval. Reinforcement treatment of a tailings dam can not only improve its stability against sliding, but also save land resources and increase the efficiency of resource use, which is of great practical significance for today’s increasingly precious land resources.

In reinforcement engineering, the interface action characteristic of reinforced soil is the key technical index [5] that directly determines the stability of the reinforcement engineering. In 1974, Schlosser and Long first used triaxial tests to study the mechanical properties of reinforced sand. From the results of triaxial compression tests, which are performed to investigate the mechanism of geosynthetics-reinforced soil, some scholars have proposed the “equivalent confining pressure principle” and “pseudo-cohesive principle” [6]. Thereafter, some scholars began to study the strength characteristics and reinforcement effect of geosynthetics-reinforced sand through triaxial tests [7,8,9]. Koerner [10] analyzed the influence of confining pressure on the shear strength of plain soil and reinforced soil through triaxial tests. Khedkar and Mandal [11] compared the stress–strain curves of pure sand, single-layer reinforced sand, and double-layer reinforced sand under three confining pressure types through triaxial tests. Khaniki and Daliri [12] conducted triaxial tests on reinforced soil and analyzed the influence of confining pressure on the shear strength index of plain soil and reinforced soil. Nair and Latha [13] determined the strength and stiffness characteristics of aggregates reinforced with geogrids with different heights, on the basis of the static and cyclic triaxial test results of a granular subbase reinforced by multi-layer geogrids. Nouri et al. [14] studied the mechanical properties of reinforced sand under triaxial monotonic drainage conditions, and determined the stress–strain and volume change characteristics and shear strength parameters of reinforced sand. Further, they estimated the strength ratio of reinforced sand at different strain levels. On the basis of the results of geogrid single reinforced triaxial test in the laboratory, Wang et al. [15] performed numerical simulations of triaxial tests by PFC3D and compared the results with those of an indoor triaxial test, analyzing the influence of reinforced layers on shear strength index and meso-parameters. The triaxial tests of geosynthetics-reinforced sandy soil conducted by these scholars focused primarily on the strength characteristics. The reinforced materials were mainly geogrids, followed by geotextiles.

For the research of reinforced tailings, at present, the interface characteristics between reinforcement and tailings are studied primarily via direct shear and pull-out tests [16,17,18]. However, the research on the interface characteristics of reinforced tailings by triaxial compression test is relatively less. Wang et al. [19] carried out uniaxial compression tests on different layers of geotextile, and studied the deformation and failure characteristics of geotextile, with the increase of the number of reinforced layers, the shear strength of geotextile increases obviously. To investigate the mechanical performance of basalt fiber-reinforced tailings (BFRT), Zheng et al. [20] carried out a series of laboratory triaxial tests, and studied the effects of five parameters (fiber length, fiber content, particle size, dry density, and confining pressure) on the mechanical properties of BFRT. Ning et al. [21] and Feng et al. [22] conducted triaxial compression tests on polypropylene woven fabric reinforced tailings with one, two, and four reinforced layers, and characterized the deformation and strength characteristics of reinforced tailings. The results showed that the existence of reinforcement mainly affected the interface parameters pseudo-cohesion, rather than the pseudo friction angle. The tailings are considered special artificial bulk sands; few comparative studies have been conducted on the reinforcement effect, the reinforcement of geosynthetics-reinforced tailings, and the existence of effective reinforcement. This paper focuses on the comparative analysis of the effect of two geosynthetics-reinforced tailings, namely geotextiles and geogrids.

To study the deformation and strength characteristics of geosynthetics-reinforced tailings, indoor triaxial compression tests of reinforced tailings with different reinforced layers were conducted. At the same time, to facilitate a comparison between the tailings, a triaxial test of plain tailings was also performed. The variation in the strength characteristics, as well as the reinforcement mechanism of geosynthetics-reinforced tailings under different reinforced layers, was explored, and the results provide a scientific basis for the structural design of geosynthetics-reinforced tailings.

## 2. Triaxial Test

To study the influence of geosynthetics-reinforced tailings on their deformation and strength characteristics, indoor triaxial tests of geosynthetics-reinforced tailings were conducted. The test device and reinforcement arrangement for the same are shown in Figure 1. Considering that the tailings structure should be in the unsaturated state under normal working conditions, and that the reinforced tailings structure is sensitive to water, a consolidated undrained unsaturated condition was adopted for the triaxial test.

### 2.1. Test Material

The tailings used in the test were taken from the tailings reservoir of Qidashan Iron Mine in Anshan, Liaoning Province, China. The tailings were grey-black, and the grain texture was relatively hard. The physical and mechanical indexes of the tailings were as follows: optimum water content 14.5%, maximum dry density 1.84 g/cm^3^, effective particle size $d_{10}=0.10\;\mathrm{mm}$, median particle size $d_{30}=0.19\;\mathrm{mm}$, restricted particle size $d_{60}=0.30\;\mathrm{mm}$, non-uniformity coefficient $C_{u}=3$, and curvature coefficient $C_{c}=1.2$. The specific particle size distribution curve is shown in Figure 2.

The geosynthetics used in the test were EGA30 glass fiber geogrid and staple fiber needle-punched geotextile. These two geosynthetics have exhibited a good application effect in various reinforcement projects. The concrete parameters of reinforcement materials are presented in Table 1.

### 2.2. Specimen Preparation

The diameter *D* and height *H* of the triaxial compression test specimens were 61.8 mm and 125 mm, respectively. To prepare the test specimens, the geogrids and geotextiles were cut into circles with a diameter of 61.8 mm. The surfaces of the reinforcement and tailings were properly scraped to increase the friction and biting effect between the reinforcement and tailings. The specimens were evenly sectioned to allow uniform spacings of single-layer, double-layer, and multi-layer reinforced specimens [21]. The specimens were compacted in six layers at an optimum water content of 14.5% and to the same degree of compaction (JTJ051-93, Test methods of soils for highway engineering) [23]. Figure 3 depicts the reinforcement layout of triaxial specimens with different reinforced layers.

### 2.3. Test Scheme and Procedure

In the test scheme, the geosynthetics (geogrids and geotextiles) were designed for the triaxial tests under four different confining pressures (100 kPa, 200 kPa, 300 kPa, 400 kPa). The test was divided into four groups: geogrids and geotextiles with 0, 1, 2, and 4 reinforced layers. A total of 28 groups of experiments were conducted. The specific scheme is detailed in Table 2.

The loading rate of the test was 1 mm/min. During the shear test, the drain valve was closed, and the test was ended when either the peak value of the principal stress difference or 15% of the axial strain was attained.

## 3. Test Results and Analysis

### 3.1. Stress–Strain Curve of Reinforced Tailings

#### 3.1.1. Different Reinforced Layers of Geogrids

The relationship between principal stress difference $\left(\sigma_{1}-\sigma_{3}\right)$ and axial strain $\varepsilon_{a}$ in the triaxial tests of geogrid-reinforced tailings under different reinforced layers is shown in Figure 4. The stress–strain curve is essentially linear at the beginning of loading. Continuous loading gradually slows the stress growth due to the increased strain. Subsequently, with the increase in axial strain, the difference in principal stress does not change significantly, and the entire loading curve shows some strain hardening characteristics. The observed characteristics indicate that the range of the triaxial test curve of geogrid-reinforced tailings increases and that the hardening characteristics of reinforced tailings become more prominent with an increasing number of reinforced layers. Additionally, the change in confining pressure is deemed to have a greater impact on the interaction between reinforcement and tailings.

#### 3.1.2. Different Reinforced Layers of Geotextiles

Figure 5 depicts the stress–strain curves of triaxial tests of geotextile-reinforced tailings under different numbers of reinforcing layers. The change in the triaxial test curve of geotextile-reinforced tailings is essentially consistent with that of geogrids, i.e., the stress–strain curve is essentially linear at the beginning of loading, and the principal stress difference reaches its peak value when the axial strain reaches approximately 10%. However, with a further increase in axial strain, the peak value of principal stress exhibits a certain downward trend, and the loading curve as a whole shows certain strain-softening characteristics. With more reinforcing layers, the range of the triaxial test curve of geogrid-reinforced tailings increases, and the softening characteristics of the test curve of reinforced tailings becomes more apparent.

### 3.2. Model Parameters of Stress–Strain Curve Fitting for Reinforced Tailings

The stress–strain curve of reinforced tailings is primarily hyperbolic, which can be fitted using the Duncan–Chang model. The fitting format is as follows:(1)$$\sigma_{1}-\sigma_{3}=\frac{\varepsilon_{a}}{a+b\varepsilon_{a}}$$
where $\sigma_{1}$ denotes the maximum principal stress, $\sigma_{3}$ denotes the minimum principal stress (confining pressure), $\left(\sigma_{1}-\sigma_{3}\right)$ denotes the principal stress difference, $\varepsilon_{a}$ denotes the axial strain, and *a*, *b* denote the experimental constant.

The relationship between $\varepsilon_{a}/(\sigma_{1}-\sigma_{3})$ and $\varepsilon_{a}$ was fitted, and is approximately linear, with *a* being the intercept and *b* the slope of a straight line. The stress–strain curves of reinforced tailings with different reinforced layers were fitted, and the fitting parameters are listed in Table 3. The fitting correlation coefficient of the stress–strain curve of reinforced tailings with different reinforced layers exceeds 90%, indicating a good fit. Therefore, we conclude that the stress–strain curves of reinforced tailings under different conditions conform to the Duncan–Chang model.

The initial tangent modulus of the hyperbola is *E_i_* = 1/*a* at the starting point of the test $\varepsilon_{a}=0$. If $\varepsilon_{a}\to \infty$, the limit deviatoric stress of the hyperbola is ${\left(\sigma_{1}-\sigma_{3}\right)}_{\mathrm{ult}}=1/b$. In the stress–strain relationship of reinforced tailings, for the case with a peak point, ${\left(\sigma_{1}-\sigma_{3}\right)}_{f}$ is the peak stress. For the case with no peak value, the deviator stress ${\left(\sigma_{1}-\sigma_{3}\right)}_{f}={\left(\sigma_{1}-\sigma_{3}\right)}_{15\%}$ is taken as $\varepsilon_{a}=15\%$. The ratio of peak stress to ultimate deviator stress is defined as the failure ratio, Refrence [24]:(2)$$R_{f}=\frac{{\left(\sigma_{1}-\sigma_{3}\right)}_{f}}{{\left(\sigma_{1}-\sigma_{3}\right)}_{\mathrm{ult}}}$$

Jabnu [25] found that the influence of confining pressure $\sigma_{3}$ on the initial shear modulus *E_i_* is expressed using the following empirical formula:(3)$$E_{i}=\mathrm{KPa}{\left(\frac{\sigma_{3}}{\mathrm{Pa}}\right)}^{n}$$
where Pa is the standard atmospheric pressure, which is 100 kPa here. *K* and *n* are the test constants, representing the intercept and slope, respectively, of the lg(*E_i_*/Pa)–lg(*σ*_3_/Pa) straight line.

The lg(*E_i_*/Pa)–lg(*σ*_3_/Pa) relationship diagram for different reinforced layers is shown in Figure 6. It is clear that all of the relationships are approximately linear, the correlation coefficient is above 85%, the intercept of each straight line is lg*K*, and the slope is *n*. The *K* and *n* values obtained are consistent within the range of *K* and *n* values of different soils [26]. The *K* values are between clay and sand, consistent with the actual situation. The relevant parameter values are summarized in Table 3. The cardinal number *K* of the initial deformation modulus *E_i_* increases with the increasing number of reinforcement layers, which indicates that the reinforcement can obviously increase the *K* value and the initial deformation modulus of tailings. *n* reflects the relationship between *E_i_* and the increase in *σ*_3_, and it increases gradually with the increasing number of reinforcement layers. This indicates that the initial deformation modulus *E_i_* of the tailings increases with increasing confining pressure.

### 3.3. Effect of Reinforced Layers on Shear Strength of Tailings

In geosynthetics-reinforced tailings, the lateral deformation of tailings is limited, which is equivalent to the confining pressure being increased. Consequently, the reinforced tailings are considered pseudo-cohesive, which can be explained using the “pseudo-cohesion principle”:(4)$$\sigma_{1}^{R}=\sigma_{3}K_{p}+2c_{R}\sqrt{K_{p}}$$
where $\sigma_{1}^{R}$ denotes the maximum principal stress of reinforced tailings under failure, $K_{p}=\tan^{2}(45{}^{\circ}+\frac{\varphi_{R}}{2})$ denotes the failure principal stress coefficient, $c_{R}$ denotes the pseudo-cohesion, and $\varphi_{R}$ denotes the pseudo friction angle.

The shear strength curves ($\sigma_{1}^{R}-\sigma_{3}$) of geogrid- and geotextile-reinforced tailings under different reinforced layers are shown in Figure 7. The shear strength curves of geogrid-reinforced tailings and geotextile-reinforced tailings are parallel to those of unreinforced tailings, indicating that it is reasonable to assume that the pseudo-friction angle is constant in the strength theory of reinforced tailings. At the same time, linear fitting of shear strength curves under different reinforced layers shows that the maximum principal stress is linearly related to the minimum principal stress with correlation coefficients greater than 90%. According to the pseudo-cohesion principle, the shear strength indexes of pseudo-cohesion and pseudo friction angle can be obtained. The concrete results are listed in Table 4.

According to Table 4, the relationship between the shear strength indexes (*c*_R_*, φ*_R_) of geogrid- and geotextile-reinforced tailings under different reinforced layers can be obtained, as shown in Figure 8. Without reinforcement, the cohesion and friction angle of the tailings are 4.91 kPa and 36.82°, respectively. For single-layer reinforcement, the pseudo-cohesion and pseudo friction angle of geogrid-reinforced tailings increased 8.74 times and 9.9% higher than those without reinforcement, whereas the corresponding values of geotextile-reinforced tailings increased 8.04 times and 7.4% higher than those without reinforcement. For double-layer reinforcement, the pseudo-cohesion and pseudo friction angle of geogrid-reinforced tailings increased to 113.1% and 1.3% higher than those of the single-layer reinforced samples, respectively, whereas the values for geotextile-reinforced tailings increased 87.9% and 3.5% higher than those of single-layer samples, respectively. When the reinforced layers were increased to four, the pseudo-cohesion of geogrid-reinforced tailings increased to 96.9% higher than that of the double-layer reinforced samples, whereas the pseudo friction angle decreased 3.3% below that for two reinforced layers. On the contrary, the pseudo-cohesion and pseudo friction angle of the geotextile-reinforced tailings increased to 110.9% and 1.3% higher, respectively, than those of their double-layer counterparts. Further analysis shows that the reinforced layers significantly affect the shear strength index of pseudo-cohesion and not the pseudo friction angle in both geogrid-reinforced tailings and geotextile-reinforced tailings. With more reinforced layers, the pseudo-cohesion of geogrids, geotextiles, and tailings are shown to increase linearly. Considering that the pseudo-cohesion plays a leading role in the interface of reinforcement and soil [27], it can be seen that compared with the reinforcement effect of geotextiles on tailings, the reinforcement effect of geogrids on tailings is more obvious, because the unique mesh structure of geogrids plays a role in the occlusion and inlay of tailings [10].

In previous research, the present author performed direct shear and pull-out tests of geogrids (used in this study) and tailings [17]. These tests were carried out under four different normal stresses (10 kPa, 20 kPa, 30 kPa, 40 kPa). Under the direct shear test conditions, the parameters of the interface between geogrids and tailings were found to be 12.11 kPa and 23.50°, respectively. The interface parameters of the grid tailings under the condition of pull-out testing were 9.33 kPa and 10.38°, respectively. Compared with the shear strength results obtained in the present study, it can be found that the pseudo-cohesion obtained by direct shear and pull-out is between the triaxial tests of plain tailings and reinforced one-layer tailings, which is considered to be caused by the lower normal stress exerted by the two tests. The comparison of shear strength results represents an added value of this research.

### 3.4. Effect Analysis of Reinforced Tailings

To evaluate the effect of different reinforced layers on the shear strength of tailings, the strength reinforcement effect coefficient $R_{\sigma }$ and the equivalent strength reinforcement effect coefficient $R_{\Delta }$ [15] are introduced. $R_{\sigma }$ represents the ratio of damage principal stress difference between reinforced tailings and pure tailings. The ratio of the increment of principal stress difference between reinforced tailings and pure tailings and the damage principal stress difference in pure tailings is characterized by $R_{\Delta }$. When $R_{\sigma }>1$ and $R_{\Delta }>0$, the reinforcement effect is significant, and a larger value corresponds with a more obvious reinforcement effect.
(5)$$R_{\sigma }={\left(\sigma_{1}-\sigma_{3}\right)}_{f}^{R}/{\left(\sigma_{1}-\sigma_{3}\right)}_{f}$$
(6)$$R_{\Delta }=\Delta {\left(\sigma_{1}-\sigma_{3}\right)}_{f}/{\left(\sigma_{1}-\sigma_{3}\right)}_{f}$$
where ${\left(\sigma_{1}-\sigma_{3}\right)}_{f}^{R}$ denotes the damage principal stress difference in reinforced tailings, ${\left(\sigma_{1}-\sigma_{3}\right)}_{f}$ denotes the damage principal stress difference in pure tailings, and $\Delta {\left(\sigma_{1}-\sigma_{3}\right)}_{f}$ denotes the increment of principal stress difference between reinforced tailings and pure tailings.

The variation rules of strength reinforcement effect coefficient and equivalent strength reinforcement effect coefficient under different reinforced layers are shown in Figure 9 and Figure 10. From these two graphs, the values of $R_{\sigma }$ and $R_{\Delta }$ are seen to increase with increasing reinforced layers. However, the rate of increase in these values is observed to decrease slowly when the reinforced layers are increased to a certain threshold, and the effect of tailings reinforcement becomes less pronounced. With a rise in confining pressure, the values of $R_{\sigma }$ and $R_{\Delta }$ decrease gradually, which indicates that the reinforcement effect of reinforced tailings triaxial specimens at low confining pressure is significant, which has a certain reference significance for the actual reinforcement of tailings engineering.

## 4. Conclusions

This paper presents an analysis of the stress–strain characteristics and shear strength performance of geogrid- and geotextile-reinforced tailings as a function of the number and type of reinforced layers, using data from triaxial tests. The major conclusions derived from the results can be summarized as follows:The stress–strain curves of geogrids show hardening characteristics, whereas those of geotextiles show softening properties. The hardening or softening characteristics become more apparent with increasing reinforced layers. The stress–strain curves of geogrid- and geotextile-reinforced tailings under different reinforced layers can be fitted by the Duncan–Zhang model.In both geogrid- and geotextile-reinforced tailings, the pseudo-cohesion of shear strength index is strongly affected by increasing reinforced layers, in which it increases linearly as the reinforced layers increase, whereas the pseudo-friction angle is affected to a lesser extent. Compared with geotextiles, the reinforcement effect of geogrids on tailings is more significant.The effect of reinforcement tailing increases with the increase of reinforcement layers, but when the number of reinforced layers increases to a certain extent, the reinforcement effect begins to weaken gradually. Reinforcement at low confining pressure can significantly improve the strength of tailings, which has a certain reference significance for the actual reinforcement of tailings engineering.

## Figures and Tables

**Figure 1 materials-13-01943-f001:**
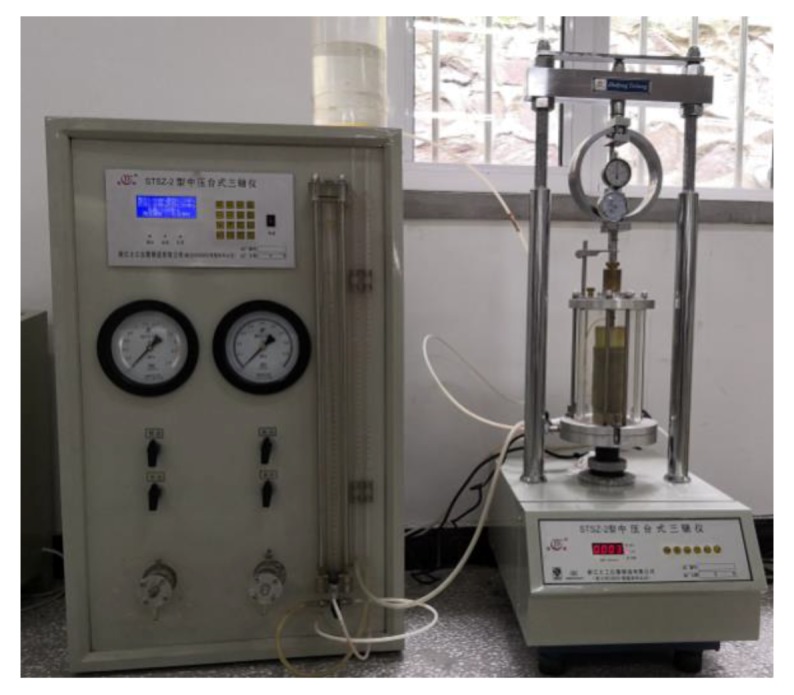
Triaxial test device.

**Figure 2 materials-13-01943-f002:**
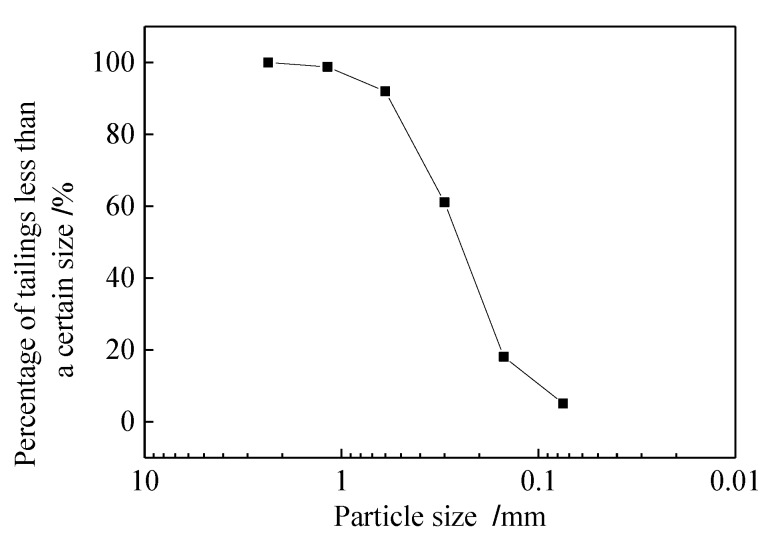
Gradation curve of tailing sand.

**Figure 3 materials-13-01943-f003:**
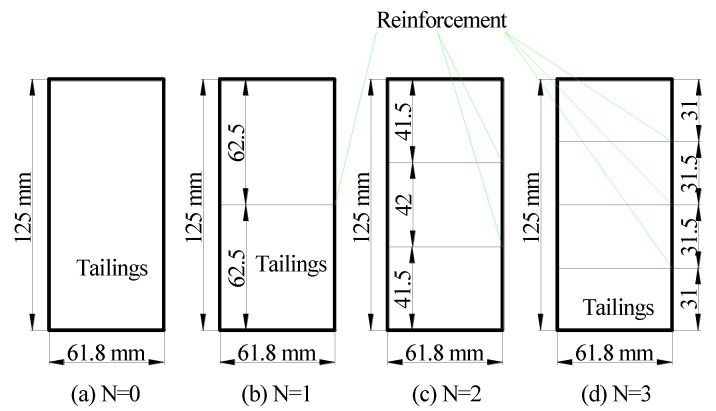
Reinforcement layout of triaxial specimens with different reinforcement layers.

**Figure 4 materials-13-01943-f004:**
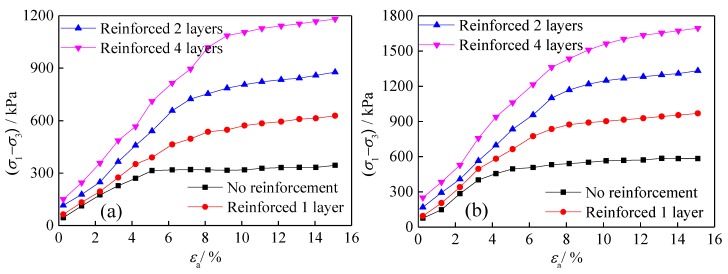
Stress–strain curves of reinforced tailings under different reinforced layers of geogrids: (**a**) Confining pressure 100 kPa, (**b**) confining pressure 200 kPa, (**c**) confining pressure 300 kPa, and (**d**) confining pressure 400 kPa.

**Figure 5 materials-13-01943-f005:**
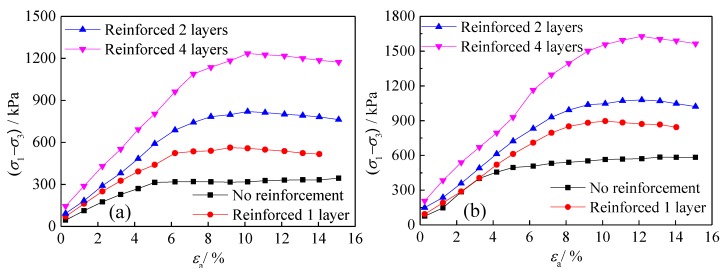
Stress–strain curves of reinforced tailings under different reinforced layers of geotextiles: (**a**) Confining pressure 100 kPa, (**b**) confining pressure 200 kPa, (**c**) confining pressure 300 kPa, and (**d**) confining pressure 400 kPa.

**Figure 6 materials-13-01943-f006:**
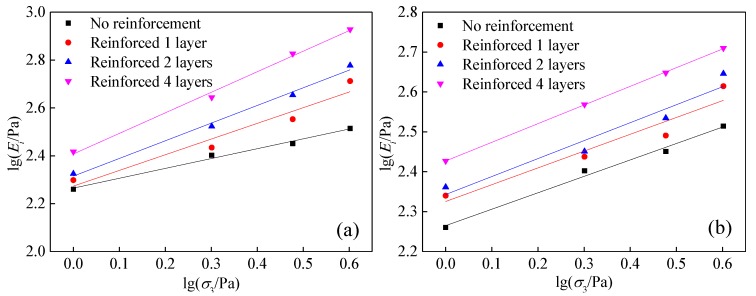
lg(*E_i_*/Pa)–lg(*σ*_3_/Pa) relation curves of different reinforced layers of tailings: (**a**) geogrids, (**b**) geotextiles.

**Figure 7 materials-13-01943-f007:**
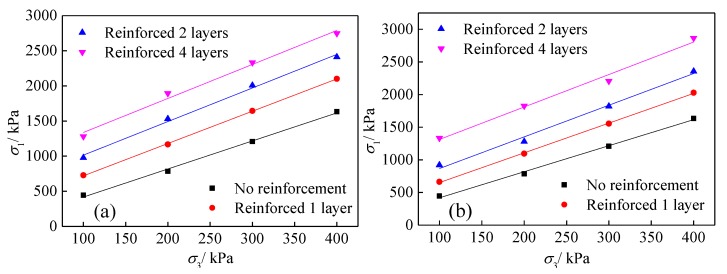
Shear strength curves of different reinforced layers: (**a**) geogrids, (**b**) geotextiles.

**Figure 8 materials-13-01943-f008:**
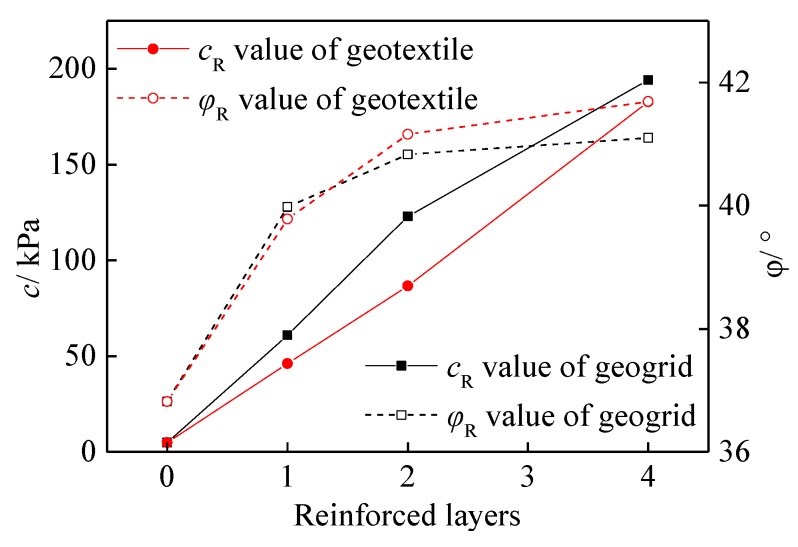
Variation in shear strength index under different reinforced layers.

**Figure 9 materials-13-01943-f009:**
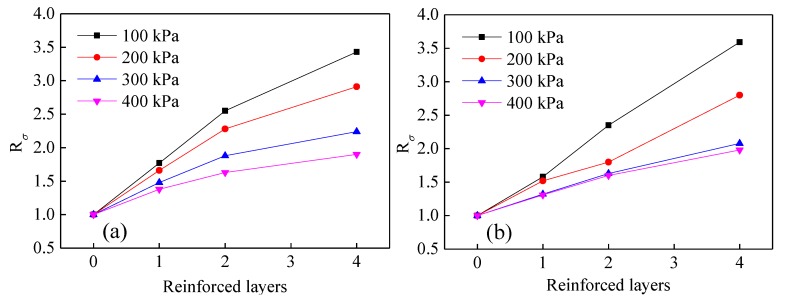
Variation in strength reinforcement effectiveness coefficient $R_{\sigma }$ under different reinforced layers: (**a**) Geogrids, (**b**) geotextiles.

**Figure 10 materials-13-01943-f010:**
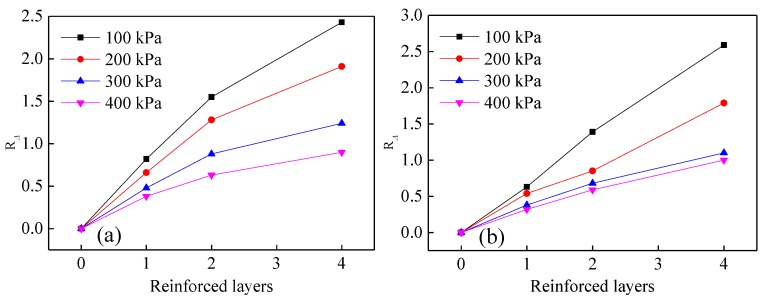
Variation in equivalent strength reinforcement effectiveness coefficient $R_{\Delta }$ under different reinforced layers: (**a**) geogrids, (**b**) geotextiles.

**Table 1 materials-13-01943-t001:** Technology parameters of geosynthetics.

EGA30 Geogrid		Technical Index	Mechanical Parameters of Needle-Punched Staple Geotextiles	Technical Index
Mesh size (length × width)/mm		12.7 × 12.7	Fracture strength of longitudinal and transverse/(kN·m^−1^)	30
Fracture strength/(kN·m^−1^)	Radial	30	Elongation corresponding to standard strength/%	40–80
	Zonal	30	CBR breaking strength ≥/kN	13
Elongation at break ≥/%	Radial	4	Tearing strength of longitudinal and transverse ≥/kN	12
	Zonal	4	Equivalent aperture/mm	0.05–0.2
Temperature resistance ≥/°C		−100 to 280	Vertical permeability coefficient/cm·s^−1^	1.0 × 10^−3^
Thickness/mm		2	Thickness/mm	2.2

**Table 2 materials-13-01943-t002:** Reinforced materials for the tests.

Test Scheme	Test Materials	Reinforced Layers	Confining Pressure/kPa	Remarks
1	No reinforcement	0	100, 200, 300, 400	Study on strength characteristics of geogrid-reinforced tailings with different reinforced layers
2	Geogrid	1		
3		2		
4		4		
5	Geotextile	1		Study on strength characteristics of geotextile-reinforced tailings under different reinforced layers
6		2		
7		4		

**Table 3 materials-13-01943-t003:** Parameters of the Duncan-Chang model with different reinforced layers of tailings.

Reinforced Layers		Confining Pressure (kPa)	*a (*×10^−5^)	*b (*×10^−4^)	*E_i_* (kPa)	(*σ*_1_ − *σ*_3_)_ult_ (kPa)	*K*	*n*	*R*_f_ Mean Value
No reinforcement		100	5.49	25.50	18,214.94	392.16	184.08	0.41	0.80
		200	3.96	14.10	25,252.53	709.22			
		300	3.54	8.16	28,248.59	1225.49			
		400	3.06	6.21	32,679.74	1610.31			
Geogrid	1	100	5.03	13.50	19,880.72	740.74	187.67	0.66	0.74
		200	3.68	7.82	27,173.91	1278.77			
		300	2.80	5.13	35,714.29	1949.32			
		400	1.94	4.53	51,546.39	2207.51			
	2	100	4.73	8.74	21,141.65	1144.16	206.59	0.74	0.72
		200	3.00	5.76	33,333.33	1736.11			
		300	2.22	4.36	45,045.05	2293.58			
		400	1.67	3.87	59,880.24	2583.98			
	4	100	3.83	5.93	26,109.66	1686.34	255.68	0.86	0.70
		200	2.27	4.33	44,052.86	2309.47			
		300	1.49	4.03	67,114.09	2481.39			
		400	1.18	2.91	84,745.76	3436.43			
Geotextiles	1	100	4.57	13.9	21,881.84	719.42	211.59	0.42	0.73
		200	3.65	8.07	27,397.26	1239.16			
		300	3.23	5.61	30,959.75	1782.53			
		400	2.43	4.25	41,152.26	2352.94			
	2	100	4.35	8.74	22,988.51	1144.16	220.29	0.45	0.69
		200	3.54	6.55	28,248.59	1526.72			
		300	2.92	4.39	34,246.58	2277.90			
		400	2.26	3.34	44,247.79	2994.01			
	4	100	3.74	5.66	26,737.97	1766.78	267.30	0.47	0.67
		200	2.70	4.16	37,037.04	2403.85			
		300	2.25	3.56	44,444.44	2808.99			
		400	1.95	2.60	51,282.05	3846.15			

**Table 4 materials-13-01943-t004:** Shear strength index of reinforced tailings under different reinforced layers.

Reinforcement	Reinforced Layers	Fitting Formula of Shear Strength	*R*^2^/%	*c*_R_/kPa	*φ*_R_/°
No reinforcement	0	$\sigma_{1}=19.60+3.99\sigma_{3}$	99.6	4.91	36.82
Geogrids	1	$\sigma_{1}=207.40+4.70\sigma_{3}$	99.2	60.97	39.98
	2	$\sigma_{1}=447.55+4.82\sigma_{3}$	99.2	122.97	40.83
	4	$\sigma_{1}=854.15+4.52\sigma_{3}$	97.2	194.13	41.10
Geotextiles	1	$\sigma_{1}=196.80+4.55\sigma_{3}$	99.9	46.12	39.78
	2	$\sigma_{1}=381.75+4.85\sigma_{3}$	98.8	86.66	41.16
	4	$\sigma_{1}=815.20+4.97\sigma_{3}$	98.3	182.78	41.69

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
