# Peer review of "Triaxial Testing of Geosynthetics Reinforced Tailings with Different Reinforced Layers"

_materials, 2020, doi:10.3390/ma13081943_

Round 1

Reviewer 1 Report

As stated by the authors, interface properties between geosynthetics and soil is an interesting topic of research. Triaxial testing of tailing material reinforced with geogrid and geotextile is presented in this paper. Fairly presented research, which could be improved significantly:

(1) Authors should argue why they have not considered matrix suction and have not performed undrained unsaturated tests as a classic unsaturated type of tests

(2) In the introduction part authors listed some previous researches, but with very low connection to presented research. The outcomes of previous researches are presented insufficiently and thus it is difficult to evaluate the novelty of presented research. Comparative studies on reinforced tailings are mentioned but with no references. The literature review could be significantly improved with more relevant references!

(3) Authors could explain how and where tested reinforced tailings are used in reality. Why they are reinforced?

(4) Four layers of reinforcement at 25 mm distance seems exaggeration

(5) Does there exist also direct shear or pull out testing of the same material? Comparison of various shear strength results could be an added value of this research.

(6) Results are analysed in the manner of Duncan-Chang model, which is not referenced. Authors should explain that model better and give the reference details.

(7) Line 175: there might be a mistake in marking of quantities in bracket.

Reviewer 2 Report

The literature review could be more extensive. It needs information about tailings structure usage. 115-166 row "When the axial strain reaches roughlu 8%, the principal stress difference basically reaches the peak value"- is it true? Please correct figure 8 title. Conclusion 3 isn't clear "On the contrary the triaxial specimen strength decreases with an increase in the reinforced layers." - is it right conclusion? Problem - scaling of geogrid properties within a regular triaxial specimen.

Reviewer 3 Report

Overall it is a good paper but it can improved.

The introduction can be modified.

why is the other soil improvement used other than geosynthetic

why is method is better

there are some research done by Murad abu Farsakh very similar to this paper, it could be great if the authors describe what is the difference between this study and that study.

Geosynthetic is widely used in reinforcement such as GRS-IBS, it could be helpful for the reader if this added to the introduction.  

Round 2

Reviewer 1 Report

The introduction part is still very poor. GRS-IB is an interesting design method for bridge support systems but has nothing to do with tailings. Do you expect bridge abutment could be constructed from tailings? If yes, then elaborate so.
• Line 38: Tailings pond is a kind of special industrial building, which is one of the three major control projects of mine [3].
Is this reference correct? Seems not.
• Reference [17] refers to elastic and permanent deformation of crushed-stone under repeated load triaxial test. I can not find it interesting for the presented paper.
I believe authors can find more references on reinforced tailings.

It can be understood from the paper that the research solves stability problem of a particular tailing dam (strengthening of tailing dam with reinforcement). Does it? Could authors briefly describe the particular dam (size etc)? What kind of reinforcement is planned?
Does the distance between reinforcement layers correspond to the reality? Analysing several layers of reinforcement and having only 25 mm spacing between them is weird. Such spacing has a sens for phenomenon research but not for modelling purpose.
